# The Predictive Role of Perceived Support from Principals and Professional Identity on Teachers’ Motivation and Well-Being: A Longitudinal Study

**DOI:** 10.3390/ijerph19116674

**Published:** 2022-05-30

**Authors:** Valérian Cece, Guillaume Martinent, Emma Guillet-Descas, Vanessa Lentillon-Kaestner

**Affiliations:** 1Teaching and Research Unit in Physical Education and Sport (UER-EPS), University of Teaching Education, State of Vaud (HEP Vaud), 1007 Lausanne, Switzerland; vanessa.lentillon-kaestner@hepl.ch; 2Laboratory of Vulnerabilities and Innovation in Sport (EA 7428), Interdisciplinary Confederation of Research in Sport (FED 4272), University of Lyon, 69100 Villeurbanne, France; guillaume.martinent@univ-lyon1.fr (G.M.); emma.guillet-descas@univ-lyon1.fr (E.G.-D.)

**Keywords:** burnout, contextual factors, self-determined motivation, teacher professional identity, vigour

## Abstract

The aim of this study was to estimate the influence of perceived support from principals and teacher professional identity (TPI) on teacher’s motivation, vigour and burnout using a longitudinal design during a school year. A sample of 544 secondary teachers reported their perceived support from principals and TPI at the beginning of the year (T1) and their self-determined motivation, vigour, and burnout both at the beginning (T1) and at the end of the year (T2). Structural equation modelling (SEM) revealed that the support from principals was associated with T1 TPI. T1 TPI only partially predicted T2 self-determined motivation (controlling T1 scores), and T2 self-determined motivation was associated with T2 burnout and vigour (controlling T1 scores). The SEM revealed a positive process involving perceived support from principals, pedagogical expertise, autonomous motivation, and well-being indicators. In summary, the present study extends the knowledge about the teacher well-being process and the role of contextual and individual antecedents. In an applied perspective, to prevent burnout, teachers need efficient initial and continuing pedagogical education to be armed in front of the students and need the support of their principals during the school year.

## 1. Introduction

The phenomenon of burnout concerns a growing number of people, including teachers. Studies have pointed out that the difficulty of the teaching profession increases the risk of teacher burnout [1,2]. Moreover, some European studies revealed percentages between 17% to 28.7% of teachers with burnout symptoms: 28.7% of teachers in Germany [3], 19.8% in Italy [4], and 17% in France (against 11% in other occupations [5]. Baeriswyl et al. [6] showed that almost 20% of Swiss teachers were at risk of burnout. The burnout phenomenon affects work quality and daily life by deteriorating the personal and social functioning of the individual. On the other hand, teachers regularly report their passion to transfer knowledge while performing a meaningful job [7]. Considering the major social importance of teacher well-being for them, the school community, and the students, the aim of this study was to explore the teacher well-being process over time. More specifically, this study aims to capture the role of the individual (i.e., teacher professional identity, TPI, and self-determined motivation) and contextual (i.e., support from principals) factors on teacher burnout and vigour across a school year.

### 1.1. Teacher Well-Being

According to Shirom et al. [8], job well-being refers to perceived physiological and psychological health. Based on the conservation of resources theory [9], well-being depends on the gain (e.g., self-esteem, equipment) or loss (e.g., reduction in salary) of resources in relation to work constraints and is directly related to psychological indices including burnout [10] and vigour [11].

#### 1.1.1. Burnout

Schaufeli and Enzmann [12] defined burnout as “a persistent, negative, work-related state of mind in “normal” individuals that is primarily characterized by exhaustion, which is accompanied by distress, a sense of reduced accomplishment, decreased motivation and the development of dysfunctional attitudes and behaviours at work” (p. 36). This definition incorporates both the process and state characteristics of burnout. Burnout has been associated with a great number of different terms such as tedium, overtraining, stress, depression, overreaching, chronic fatigue, staleness [12], but it has been clearly differentiated from these symptoms [12]. Teacher burnout is related to different negative outcomes such as absenteeism, turnover [13], or teacher mental ill-health [14]. Moreover, burnout has been associated with low levels of student academic achievement and poorer student motivation [15].

To assess professional burnout in general, Maslach and her colleagues constructed the Maslach Burnout Inventory–General Survey (MBI-GS [16]) with three subscales, namely exhaustion, cynicism, and reduced personal efficacy. Even if the burnout conceptualization by Maslach and colleagues is the most used to date, this conceptualization and its measurement tool have been criticized. Some studies found that emotional exhaustion and depersonalization subscales tended to collapse into one factor [17]. Shirom and Melamed [18] argued that the three burnout dimensions were not theoretically deducted but resulted from labelling exploratory factor-analysed items initially collected to reflect the range of experiences associated with the burnout phenomenon. As a result, an alternative conceptual approach was developed related to feelings of physical, emotional, and cognitive exhaustion [10]. This conceptual approach has led to the construction of the Shirom-Melamed Burnout Measure (SMBM). Theoretically, the SMBM is based on Hobfoll’s Conservation of Resources (COR) [9] theory and is related to energetic resources only. COR theory was applied in several articles to conceptualize burnout [19,20]. The SMBM is organized around three dimensions: physical fatigue, emotional exhaustion, and cognitive weariness. Physical fatigue shows a facet of burnout, spotted clinically [21]. Emotional exhaustion is the most robust dimension of MBI [22]. Finally, the third dimension, cognitive weariness related to difficulties to focus and quickly mobilize intellectual abilities. The SMBM has the potential of revealing more information about the burnout process than the MBI-GS and focuses particularly on the core content of burnout—physical, emotional, and cognitive exhaustion. In addition, the varying meanings of burnout as assessed by the SMBM in a different job and occupational categories appear relevant to elucidate the different pathways linking burnout with aspects of physical health and consequently seem better adapted for our multidisciplinary project.

#### 1.1.2. Vigour

Maslach and Leiter [23] considered burnout and work engagement to be the opposite poles of a continuum: “Energy, involvement, and efficacy are the direct opposites of the three dimensions of burnout” (p. 34). Contrary, Schaufeli and Bakker [24] did not consider that engagement is adequately measured by the opposite profile of MBI scores. For instance, feeling emotionally drained from one’s work “once a week” does by no means exclude that in the same week one might feel bursting with energy. From this perspective, burnout and engagement were considered independent states and negatively related. They defined engagement as a positive, fulfilling, work-related state of mind that is characterized by vigour, dedication, and absorption [25]. However, Shirom [11] highlighted concerns about the definition of vigour presented by Schaufeli et al. [25]. Indeed, he emphasized that this conceptualization overlaps with other psychological constructs such as psychological persistence [26], job involvement [27], and resiliency [25]. He also highlighted that the three dimensions of vigour came from a data- instead of a theory-driven approach. To overcome these limitations, Shirom [11] defined vigour as a positive multidimensional affect with three dimensions: the sensation of physical strength (i.e., individual own physical abilities), emotional energy (i.e., individual ability to express sympathy and empathy toward others), and cognitive liveliness (i.e., individual thinking skills and mental agility). A questionnaire was developed (i.e., Shirom-Melamed Vigor Measure; SMVM [28]) measuring these three vigor components. According to Shirom [29], professional vigour is related to many adaptive outcomes including job satisfaction and work achievement.

Thus, the major effects of teacher well-being encourage exploration of the entire burnout and vigour process for teachers. In this perspective, previous empirical and theoretical studies in the work domain provide potential predictors, including both personal (e.g., self-determined motivation, TPI) and contextual factors such as support from principals [30].

### 1.2. The Role of Self-Determined Motivation

Self Determination Theory (SDT) postulates that motivation to engage in specific behaviours can be situated on a continuum ranging from controlled to autonomous motivation, with autonomous motivation reflecting a higher quality of motivation than controlled motivation [31]. SDT identifies autonomous and controlled motivation as qualitatively different forms of motivation. Controlled motivation refers to feeling externally (e.g., preparing lessons because of a school/director inspection) or internally (e.g., proving oneself and showing off own skills as a good teacher) pressured or coerced to engage in specific behaviours or activities. Autonomous motivation involves a sense of volition (e.g., enjoying enriching students with new insights and knowledge) and self-endorsement (e.g., valuing the importance of transferring certain skills to students). In most studies on antecedents and outcomes of teacher motivation quality to teach, a variable-centred approach has been taken. In these studies, autonomous motivation related to more optimal outcomes, such as more commitment and vigour in the work setting [32], while controlled motivation related to more negative outcomes, such as burnout [33]. Numerous studies have shown the protective role of self-determination in the well-being processes among teachers [34,35,36].

The effect of environmental demands is particularly harmful to the teachers’ psychological well-being when they perceive that their self-determination is threatened [34]. A personality variable, such as teacher professional identity (TPI), may also indirectly have an influence on teacher well-being by moderating the relationship between teaching context and teacher motivation.

### 1.3. The Role of Teacher Professional Identity

The concept of TPI has gained considerable attention these last years. As underlined by Beijaard et al. [37], “in most studies, the concept of professional identity was defined differently or not defined at all.” (p. 125). The lack of consensus around TPI definition and validated tools has reduced, until recently, the possibilities to build a solid theoretical framework around TPI [38,39,40] and it “has led researchers to try to identify major components that constitute TPI in relation to particular research emphases” [40]. The definition of TPI is complex, but there is a general acknowledgment of its multifaceted and dynamic nature [39,41]. Beijaard et al. [37] underlined the four following features as essential for TPI: (a) it is an ongoing process of interpretation and re-interpretation of experiences, (b) it implies both person and context, (c) it consists of sub-identities that more or less harmonize, and (d) the agency, meaning that teachers have to be active in this process. TPI can be conceptualized as a self-definition as a teacher. TPI can also be seen as an answer to the following questions: “Who am I at this moment?” [37] and “How do I see my role as a teacher?” [42]. More precisely, in this study, the definition of TPI is based on teachers’ perception of expertise in relation to teacher roles such as in previous studies [39,42,43,44,45]. A French-language questionnaire (Questionnaire on Perceived Professional Identity among Teachers) was recently validated to measure personal components of TPI based on teachers’ perceptions of expertise [43]. More precisely, it allows measuring two domains of teaching expertise in perceived TPI, one directly related to teaching contents (i.e., subject matter expertise) and the other related to the teacher-students relationships (i.e., pedagogical expertise).

Beijaard et al. [37] underlined that positive self-perception of TPI appeared to override his or her dissatisfaction with poor working conditions. Rascle and Bergugnat [46] showed that the more the teachers considered the profession painful, the more they are at risk of emotional exhaustion. In addition, in the school context, teacher identity has been recognised as an important factor influencing how a teacher teaches [37].

Moreover, as mentioned by Roth [47], “workers’ sense of autonomy is not only a function of their current context. It also incorporates long-term developmental processes of personal integration and identity development.”. TPI considered as teaching expertise [38], is closely related to teacher psychological experiences [41]. In particular, TPI has been associated with teachers’ perspectives and job motivation [48].

### 1.4. Teachers’ Perceived Support from Principals

In research assessing the impact of work environment factors on teacher burnout, different types of roles, interactions, demands, and resources can vary considerably in each context and as a result, can have a differential influence on teacher psychological experiences [49,50,51,52]. Many school-level factors were identified as related to teacher well-being indicators, and especially burnout, ranging from the quality of relationships with parents, colleagues, and supervisors to the adequacy of school resources. This study focused on the relationship with the support from school principals who are responsible for the school policies and hierarchical management and have been identified as one of the major actors in the teacher professional experiences [53,54]. Moreover, Friedman’s [55] comparison of the organisational characteristics of “high burnout” schools in Israel (those in which teachers reported high levels of burnout) and “low burnout” schools demonstrated the importance of school climate factors such as supervisor support, teacher autonomy and work pressure on teacher development. High burnout schools typically had a rigid management structure where decisions came from the principal and teachers did not work as a team. In contrast, low burnout schools typically had much looser structures where teachers could regularly speak to the principal and contribute to decision-making within the school. Numerous studies suggested direct relationships between school environment factors and mental health indicators [51,56,57,58]. However, other studies suggested more moderate relationships. Byrne [49], for example, found that school-level factors including social support and participation in decision-making played an important but indirect role in well-being development. Finally, previous studies highlighted the role of school contextual factors on TPI [59]. In particular, the quality of the relationships within the school community seems strongly associated with teachers’ self-perception and perceived expertise [59]. Moreover, several studies have highlighted the significant role of the principals on teacher development by the implementation of a specific learning environment in the school or an instructional leader role [53,54,60].

Thus, considering the significant role of principals in teacher development, motivation, and well-being [49,53,55], it seems crucial to explore the role of the perceived support from principals on TPI and teachers’ psychological experiences. From an applied perspective, the investigation of the role of the principals on teacher well-being could promote interesting practical implementations focused on leadership style.

### 1.5. Study Relevance and Purpose

Studies on teacher well-being have often prioritized a single time of measurement, assuming that burnout is a stable process over time [34]. Yet some studies have shown variations over time in the burnout process, especially among teachers [34,36,57,61]. The literature in this field posits that professional well-being is related to both personal and contextual resources [30]. In particular, teacher well-being has been closely related to motivational factors [36] and self-perception [48]. Moreover, burnout and vigour also depend on contextual resources and social support, especially including support from principals for teachers considering their role in teacher development [49,53]. However, the associations between these variables have been mainly investigated using a cross-sectional design. Thus, the aim of this study was to estimate the role of perceived support from principals and TPI on teachers’ motivation, vigour, and burnout using a longitudinal design during one school year. Based on both theoretical [10,11,57,61] and empirical studies [36,48,52], we assessed that the perceived support from principals would be positively associated with the scores of TPI (Hypothesis 1). We also predicted that the levels of TPI at the beginning of the scholar year predict the self-determined motivation at the end of the year (Hypothesis 2). Finally, we expected self-determined motivation associated with well-being indicators (Hypothesis 3). Specifically, the scores of autonomous motivation would be positively associated with vigour and negatively with burnout levels, whereas controlled motivation and amotivation would be negatively associated with vigour and positively with burnout levels.

## 2. Materials and Methods

### 2.1. Participants

A sample of 544 secondary teachers (Mean age = 42.16 (±9.59) years) with 16.36 years of experience on average (±9.46) including 350 French teachers and 194 teachers from the French-speaking part of Switzerland participated in the present study. A total of 188 teachers were male and 326 were female. There were thirty teachers who did not report their sex. The teachers taught different school disciplines such as physical education, mathematics, history, French, English, German, physics, biology, and economy.

### 2.2. Procedure

The research was conducted between September 2018 and July 2019, in accordance with the principles of international ethical guidelines. Permission to conduct the study was granted by the ethics board of the host universities. The teachers were contacted using the institutional mailing list including all the permanent and public secondary teachers of the states of Vaud (Switzerland) and the region Auvergne Rhône Alpes (France). Teachers were reminded that their participation was voluntary, that their responses were confidential and that they could withdraw from the study at any time. Support from principals and TPI were assessed only at the beginning of the scholar year (Time 1, T1; from September to October) whereas the scores of work motivation, burnout and vigour were measured at both the beginning and the end of the year (Time 2, T2; from May to July). The questionnaires were completed online. Participants provided written informed consent by e-mail. They needed 15 min to complete the questionnaire.

### 2.3. Measures

#### 2.3.1. Burnout and Vigour

The French version [62] of the Shirom-Melamed Burnout Measure (SMBM [28]) was used to assess burnout levels. A total of fourteen items and three subscales were used to measure physical fatigue (α = 0.92 and 0.93 at T1 and T2 respectively; 6 items), cognitive weariness (α = 0.92 and 0.93 at T1 and T2; 5 items), and emotional exhaustion (α = 0.74 and 0.75 at T1 and T2; 3 items).

Vigour was measured using the French version [63] of the Shirom-Melamed Vigour Measure (SMVM [11]). Three subscales and 12 items were used to measure physical strength (α = 0.94 and 0.95 at T1 and T2 respectively; 5 items), cognitive liveliness (α = 0.82 and 0.85 at T1 and T2; 3 items), and emotional energy (α = 0.86 and 0.89 at T1 and T2; 4 items). For these two scales, the participants responded on a 7-point Likert scale with values ranging from 1 (never) to 7 (always).

#### 2.3.2. Work Motivation

The teachers completed the French version of the Multidimensional Work Motivation Scale (MWMS [64]). The 19–item questionnaire measures three factors: amotivation (α = 0.77 and 0.72 at T1 and T2 respectively; 3 items), controlled motivation (α = 0.80 and 0.81 at T1 and T2, 10 items including external and introjected regulation), and autonomous motivation (α = 0.85 at T1 and T2; 6 items including identified regulation and intrinsic motivation). The participants responded on a 7-point Likert scale with values ranging from 1 (not at all) to 7 (completely).

#### 2.3.3. Teacher Professional Identity

The QIPPE scale [43] was used to assess TPI. This 11-item scale includes two individual components of TPI: pedagogical expertise (T1 α = 0.75; 6 items) and subject matter expertise (T1 α = 0.76; 5 items). Teachers rated each item on a 5-point Likert scale ranging from 1 (never) to 5 (always).

#### 2.3.4. Support from Principals

The 6-item version of the Work Climate Questionnaire (WCQ [65]) was used to assess the support from principals, self-perceived by the teachers (T1, α = 0.86). The questions are stated with respect to the support of the school principals. The teachers were asked how much the principals understand (e.g., “I feel understood by my principal”), communicate confidence to teachers (e.g., “My principal communicates his/her confidence in my work”), or offer and accept teachers’ choices (e.g., “I feel that my principal give me choices and options”). Teachers rated each item on a 5-point Likert scale ranging from 1 (never) to 5 (always).

### 2.4. Data Analyses

All the analyses were conducted using Mplus Version 7.3 (Los Angeles, CA, USA). All models were estimated using the robust maximum likelihood estimator (MLR) to account for the potential multivariate non-normality of the data [66].

First, the cultural invariance between French and Swiss teachers was tested to ensure the questionnaires have the same meaning despite the cultural and contextual differences, such as work time or the number of students per class [67]. Following van de Schoot et al.’s instructions [68], we tested configural (no equality constraints), metric (equal item loadings), scalar (equal item loadings and item intercept concurrently), residual variance (equal error variance), residual covariance (equal error covariance), factor variance (equal factor variance) and factor mean invariance (equal factor mean). The difference between two nested models was determined by differences in CFI and RMSEA values. A change of less than 0.010 in CFI, 0.015 and in RMSEA provided evidence of cultural invariance [69].

Then, the relationships between the variables were investigated using structural equation modelling (SEM). Full information maximum likelihood was used to handle missing data by estimating a likelihood function for each participant based on the present variables. SEM refers to statistical models which allow exploration of the relationships between the variables, including all the latent variables in the same model. This approach prevents type I errors in comparison to independent tests of each relationship.

In line with Anderson and Gerbing [70], we conducted the two-step modelling procedure composed of (1) the measurement model and (2) the structural model. For the measurement model step, we computed both a correlated model and a correlated model in which factor loadings of each indicator were constrained to be equal across time. Items were averaged to create three indicators (parcels [71]) of autonomous motivation, controlled motivation, pedagogical expertise, subject matter expertise, physical fatigue, cognitive weariness, emotional exhaustion, physical strength, cognitive liveliness, and emotional energy. For instance, the first and the second items of autonomous motivation (i.e., intrinsic motivation and identified regulation) were averaged to create one of the three parcels. The parcels were the same for T1 and T2. We used a combination of indices to achieve a comprehensive evaluation of fit [72] including the chi-square (χ^2^), the Comparative fit index (CFI), the Tucker-Lewis index (TLI), the root mean square error of approximation (RMSEA) and its confidence interval (90% CI). CFI and TLI of 0.90 and 0.95 are taken to reflect acceptable and excellent fits, respectively, whereas RMSEA of less than 0.06 and 0.08 is taken to reflect close and reasonable fits, respectively [68]. Akaike’s information criterion (AIC), Bayesian information criteria (BIC) and sample-size adjusted BIC (ABIC) were used for comparison with alternative models and provided an indication of which model yields the better fit to the data [73].

We then tested the longitudinal relationships between T1 support from principals and TPI, T2 self-determined motivation, burnout, and vigour, controlling for T1 scores (see Figure 1).

In line with our hypothetical model, we tested a model where (a) T1 support from principals predicts T1 TPI; (b) T1 TPI predicts T2 self-determined motivation (controlling scores of T1 self-determined motivation); (c) T2 self-determined motivation predicts T2 burnout and vigour (controlling scores of T1 self-determined motivation, burnout, and vigour). Two later models were performed to test the moderating effect of sex (male vs. female) in the relationships. Because no difference was observed in the significant relationships between the two models, only the general model was presented in this study.

## 3. Results

### 3.1. Preliminary Analyses

The descriptive statistics and correlation matrix are presented in Table 1. The goodness-of-fit indices of the several steps for cultural invariance are presented in Table 2. The configural invariance model (i.e., without any constraints) revealed an acceptable fit to the data. Model fit was still acceptable when invariance constraints for metric, scalar, residual variance, residual covariance, factor variance, and factor mean invariance (CFI = 0.93 to 0.93; TLI = 0.92 to 0.92; RMSEA = 0.052 to 0.054). Moreover, ΔCFI (<0.010), and ΔRMSEA (<0.015) provided evidence of cultural invariance of the present data. Finally, 87 teachers answered the questionnaire at T1 but not at T2. To inspect the possibility of sample bias, we computed a series of analyses of variance to explore whether teachers who withdrew from the study between the two times points differed from those who responded at both times. 

MANOVAs did not reveal significant differences concerning the scores of the dependent variables (*F*(1) < 1.34, *p* > 0.05).

### 3.2. Measurement Model

First, the correlated model (M1) and the correlated model in which factor loadings of each indicator were constrained to be equal across time (M2) provided an acceptable fit to the data (CFI = 0.94, TLI = 0.93, RMSEA = 0.034). Hence, the equality constraints imposed on factor loadings across time did not affect the overall fit of the model. The fit indices provided evidence for the relative similarity of M1 and M2. Thus, because time invariance in factor loadings was required to ensure that the latent variables were the same at each time point, M2 was retained for the subsequent analyses.

### 3.3. Standardized Estimates for the Structural Model

The results of the relationships between the variables are summarized in Table 3 and Table 4, and the significant results of the hypothetical models are highlighted in Figure 1. T1 pedagogical and subject matter expertise were significantly associated with T1 perceived support from principals (*β* = 0.24, *p* < 0.01 and *β* = 0.12, *p* < 0.05 respectively). In other terms, the more the teachers consider that their principals support them, the more they reported high levels of professional expertise. T2 autonomous motivation was marginally significantly and positively predicted by T1 pedagogical expertise (*β* = 0.54, *p* = 0.08). The other scores of T2 self-determined motivation were not significantly predicted by the two components of T1 TPI. Finally, as can be expected, the scores of T2 amotivation, autonomous, and controlled motivation were significantly predicted by their T1 scores (*p* < 0.01).

## 4. Discussion

The aim of this study was to capture the influence of perceived support from principals and TPI on secondary teachers’ motivation, vigour, and burnout using a longitudinal design during one school year. Based on both theoretical frameworks [10,11,31,59] and empirical studies [36,48,52], we hypothesized that perceived support from principals would be positively associated with the scores of TPI. We also predicted that the levels of TPI at the beginning of the scholar year predict the self-determined motivation at the end of the year (controlling self-determined motivation at the beginning of the year). Finally, we expected that self-determined motivation would be associated with well-being indicators. First, the preliminary analyses confirmed the cultural invariance between French and Swiss teachers. Thus, despite cultural teaching differences [67], the questionnaires have the same meaning for the two samples. Then, the SEM provided adequate fit indices for the hypothetical model and allowed exploration of the relationships between the selected latent variables.

In line with our hypothetical model (Hypothesis 3), the scores of autonomous motivation were negatively associated with the three burnout dimensions and positively related to emotional energy and physical strength, whereas the scores of amotivation were negatively related to the levels of vigour, and positively associated with the levels of burnout. These results are consistent with SDT [31,32] and with previous studies driven by SDT in a work domain [35,36]. Specifically, in line with Gagné and Deci [32], this finding confirms the protective role of autonomous motivation on teacher burnout and highlights the associations between motivation and professional adjustment in front of school constraints across time. Following a COR perspective [9,11], the relationships between the self-determined motivation and the well-being indicators suggest that a good quality of motivation allows conservation of the personal resources despite the school constraints, during the school year.

However, the scores of controlled motivation were sparsely related to burnout and vigour in the present results. This result is not in line with the tenets of SDT which postulate that controlled motivation is particularly related to maladaptive outcomes [32,64]. However, the valence of this form of motivation is unclear in the short term [74]. Thus, further studies should explore the long-term effects of the experiences of controlled motivation for teachers to complete the present findings. Finally, controlled motivation seems only related to one component of vigour. This result confirms the suitability to investigate both burnout and vigour in a bivariate approach and confirms the separate processes for the two outcomes [11].

The SEM analyses allowed us to explore the longitudinal relationships between TPI and self-determined motivation. In line with SDT and our hypotheses (Hypothesis 2), autonomous motivation at the end of the scholar year was marginally predicted by the scores of pedagogical expertise at the beginning of the year. This result is in line with previous similar results [39,41] which revealed that high TPI was associated with positive psychological outcomes. Moreover, this relationship extends the associations between motivation and TPI highlighted by Richardson and Watt [48]. This positive relationship suggests that teachers with high levels of pedagogical expertise adapt to the motivational constraints during the school year. In particular, we could hypothesize that the pedagogical expertise is related to the scores of autonomous motivation because of a better feeling of competence, which refers to a theoretical nutriment of self-determined motivation.

However, the subject matter expertise was not related to motivation scores whereas controlled motivation and amotivation at the end of the scholar year were not predicted by TPI scores. This absence of a relationship suggests that teacher well-being is more dependent on pedagogical factors than subject matter expertise. Another assumption for this result is that subject-matter expertise is entirely learned, whereas pedagogical expertise is also closely related to intimate factors [43] and thus more associated with psychological states. Moreover, the more maladaptive forms of motivation (controlled motivation and amotivation) were not predicted by T1 TPI. Thus, controlled motivation and amotivation could be more associated with the social environment than individual factors. In summary, the present results suggest that TPI only partially and marginally predicts the levels of teacher well-being.

Another aim of this study was to explore the role of the support from principals on TPI. In line with our hypotheses and previous studies revealing the role of school leaders on teacher development [49,53,54,60], the support from principals was positively associated with both pedagogical and subject matter expertise (Hypothesis 1). The positive relationships confirm the leading role of the school context on individual perceptions and TPI. As mentioned by TPI literature [39,59], teachers develop their identity depending on their professional context and their interactions with the learning community. Specifically, in line with the study of Bredesson [53], this result confirms that the support from principals is associated with the level of teacher professional development. This association suggests that supportive principals are able to promote teachers’ self-perception of professional expertise providing opportunities for choices, giving compliments and value to teachers’ work, using positive interactions, and including the teachers in school functioning and pedagogical policies. This finding extends the previous investigation of TPI process and highlights the role of this antecedent in teachers’ self-perception.

Moreover, considering the previous successive relationships mentioned in the present study (i.e., between TPI and autonomous motivation, and between autonomous motivation and well-being indicators), these significant relationships revealed the indirect role of the support from principals on teacher well-being including burnout and vigour [49,52].

### 4.1. Implications for Practice

From a practical perspective, the present results could have implications at different levels. The SEM relationships provide a virtuous process involving successively, the support from principals, the pedagogical expertise, the autonomous motivation, and a high vigour or low burnout scores. These positive associations are in line with the tenets of positive psychology [75] highlighting the importance to focus on adaptive constructs. Firstly, concerning the individual factors, the present results encourage the support of the professional identity and the self-determined motivation of teachers to promote well-being. It seems interesting to implement school programs focused on teachers’ motivation. Driven by SDT, motivational programs could promote teachers’ self-determined motivation through initial and continuing training [47,76]. For instance, during teacher training, it seems interesting to invite them to make personal choices to support the need for autonomy [77]. Similarly, from a health perspective, the present results highlight the importance of pedagogical expertise and social relationships with students during teacher training. Moreover, on the contextual dimension, the positive associations between perceived support from principals and TPI confirm the indirect effect of the environment on teacher well-being. Consequently, it appears crucial to alert the whole school community about the risks and benefits related to the principals’ behaviours and management. In a work context, interventional studies have confirmed the promising effect of interventions given to managerial hierarchy on workers’ motivation, well-being, and development [78,79]. Thus, it seems interesting to include the principals in programs focused on teacher development by implementing workshops highlighting the benefits of social support and providing strategies to offer choices or boost teacher confidence. In summary, the combination of findings of the present study incites promotion of the teachers’ self-perception of competence and to value teachers from a contextual and individual perspective.

### 4.2. Limits and Perspectives

Different limitations of this study should be mentioned. First, the present study was driven by a data-centred approach. However, the teachers’ well-being is multifactorial, and it could be interesting to explore the role of the subject taught (e.g., physical education vs. other disciplines [67,80]). Similarly, other moderators could be integrated into the contextual part of this process. Especially, the teaching socio-economical context (e.g., rough vs. privileged environment) or the school level (e.g., primary vs. secondary school level) could have an impact on the selected variables and should be included in the models of further studies. Other mediated measures could also improve the model. In line with SDT, it should be more accurate to include the basic psychological needs satisfaction and thwarting. The well-being indices should be completed with qualitative and objective measures such as interviews, physiological and neuro-physiological data [81]. Second, the present study used a two-wave longitudinal design. This approach could be limited to capture the teachers’ psychological process. Future similar studies should use a more intensive data collection for a better understanding of the process variability. Moreover, a teaching career refers to successive periods including reflection, renewal, and growth cycles [82]. Thus, more relationships between variables (e.g., between TPI and self-determined motivation) would emerge from future studies exploring the process for several years. Considering the significant changes in the learning styles (e.g., development of online instructions) [83] and teachers’ psychological experience [80] driven by the COVID pandemic, it seems to be relevant to investigate teacher experiences after the crisis to explore the potential differences in the teachers’ well-being process. Finally, the present study is limited to measures focused on teachers. Considering the social impact of teacher well-being on students’ achievement, further studies should include the students with measurements of students’ self-determined motivation, well-being, and academic performance. Finally, while the present results highlight the role of the perceived support from principals in the teachers’ experiences, further studies should include effective measures of the management style of principals.

## 5. Conclusions

To conclude, this study highlighted the successive relationships between perceived support from principals, TPI, teachers’ self-determined motivation, burnout, and vigour across a scholar year. The SEM revealed a positive process involving perceived support from principals, pedagogical expertise, autonomous motivation, and well-being indicators. Specifically, TPI (both pedagogical and subject-matter expertise) was associated with the support from principals, whereas only pedagogical expertise predicted the scores of autonomous motivation (but not controlled motivation and amotivation) at the end of the scholar year. Finally, the present results confirmed the major role of self-determined motivation on teacher burnout and vigour. In sum, the present longitudinal study highlights the role of both individual (TPI) and contextual factors (support from principals) in teacher motivation and well-being during the school year. From an applied perspective, to prevent burnout, secondary teachers need efficient initial and continuing pedagogical education to be armed in front of the students and need the support of their principals during the school year. In the future, it will be necessary to continue to explore the role of contextual (school collaboration for instance) and individual variables (TPI especially), and their mediated or moderated function in teacher well-being.

## Figures and Tables

**Figure 1 ijerph-19-06674-f001:**
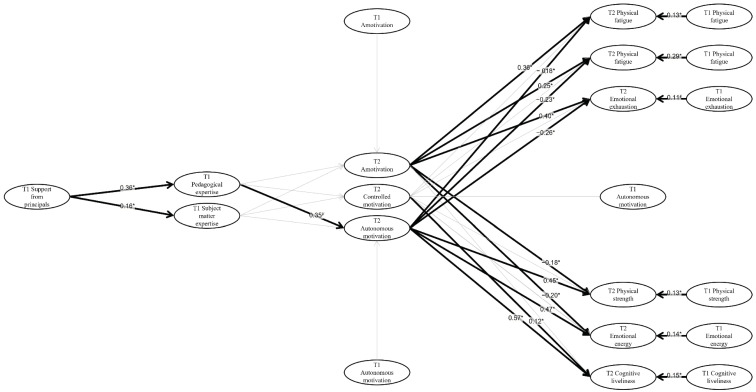
Significant relationships between the study variables. Note. * *p* < 0.05; ^¥^ *p* < 0.10. The scores of the three dimensions of T2 burnout were positively and significantly related to T2 amotivation (*β* = 0.64, *p* < 0.01 for physical fatigue, *β* = 0.39, *p* < 0.01 for cognitive weariness, and *β* = 0.40, *p* < 0.01 for emotional exhaustion) and negatively related to T2 autonomous motivation (*β* = −0.27, *p* < 0.05 for physical fatigue, *β* = −0.30, *p* < 0.01 for cognitive weariness, and *β* = −0.21, *p* < 0.01 for emotional exhaustion). In contrast, the scores of T2 vigour were negatively and significantly related to T2 amotivation (*β* = −0.27, *p* < 0.05 for physical strength, and *β* = −0.27, *p* < 0.05 for emotional energy) and positively by T2 autonomous motivation (*β* = 0.56, *p* < 0.01 for physical strength, *β* = 0.62, *p* < 0.01 for cognitive liveliness, and *β* = 0.53, *p* < 0.01 for emotional energy). In addition, T2 cognitive weariness was marginally significantly and positively predicted by T2 controlled motivation (*β* = 0.09, *p* = 0.06). Finally, the scores of T2 burnout and vigour were significantly predicted by their T1 scores (*p* < 0.01).

**Table 1 ijerph-19-06674-t001:** Means, Standard Deviations, and Correlations of the Variables.

	1	2	3	4	5	6	7	8	9	10	11	12	13	14	15	16	17	18	19	20	21
1. T1 Support from principals																					
2. T1 Pedagogical expertise	0.08																				
3. T1 Didactical expertise	0.01	0.38 *																			
4. T1 Amotivation	0.26 *	0.18 *	0.23 *																		
5. T1 Controlled motivation	0.30 *	0.05	0.11	0.23 *																	
6. T1 Autonomous motivation	0.30 *	0.37 *	0.36 *	0.57 *	0.36 *																
7. T1 Physical fatigue	0.24 *	0.15 *	0.12	0.35 *	0.01 *	0.38 *															
8. T1 Cognitive weariness	0.17 *	0.22 *	0.26 *	0.23 *	0.04 *	0.30 *	0.59 *														
9. T1 Emotional exhaustion	0.16 *	0.30 *	0.18 *	0.21 *	0.07 *	0.31 *	0.41 *	0.54 *													
10. T1 Physical strength	0.29 *	0.31 *	0.32 *	0.43 *	0.12 *	0.48 *	0.68 *	0.54 *	0.38 *												
11. T1 Emotional energy	0.28 *	0.46 *	0.29 *	0.36 *	0.09 *	0.46 *	0.36 *	0.39 *	0.58 *	0.57 *											
12. T1 Cognitive liveliness	0.27 *	0.33 *	0.48 *	0.30 *	0.05 *	0.47 *	0.39 *	0.52 *	0.34 *	0.69 *	0.61 *										
13. T2 Amotivation	0.09	0.12	0.08	0.14 *	0.02	0.08	0.04	0.15 *	0.08	0.09	0.02	0.03									
14. T2 Controlled motivation	0.03	0.00	0.06	0.03	0.15 *	0.00	0.15 *	0.12	0.04	0.05	0.04	0.12	0.15 *								
15. T2 Autonomous motivation	0.13 *	0.12	0.01	0.12	0.04	0.05	0.02	0.10	0.00	0.07	0.03	0.01	0.60 *	0.31 *							
16. T2 Physical fatigue	0.11	0.05	0.00	0.05	0.01	0.07	0.21 *	0.26 *	0.14 *	0.12	0.06	0.11	0.39 *	0.05	0.35 *						
17. T2 Cognitive weariness	0.10	0.10	0.07	0.10	0.03	0.08	0.28 *	0.40 *	0.26 *	0.20 *	0.17 *	0.19 *	0.33 *	0.01	0.31 *	0.73 *					
18. T2 Emotional exhaustion	0.04	0.03	0.06	0.05	0.06	0.02	0.12 *	0.19 *	0.17 *	0.09	0.06	0.09	0.43 *	0.00	0.40 *	0.59 *	0.61 *				
19. T2 Physical strength	0.14 *	0.17 *	0.09	0.01	0.10	0.09	0.13 *	0.25 *	0.09	0.15 *	0.07	0.15 *	0.40 *	0.19 *	0.50 *	0.67 *	0.61 *	0.52 *			
20. T2 Emotional energy	0.08	0.12 *	0.10	0.03	0.13 *	0.04	0.01	0.16 *	0.05	0.10	0.07	0.13 *	0.42 *	0.11	0.50 *	0.36 *	0.37 *	0.63 *	0.58 *		
21. T2 Cognitive liveliness	0.10	0.20 *	0.14 *	0.10	0.09	0.11	0.12 *	0.24 *	0.08	0.17 *	0.10	0.16 *	0.37 *	0.18 *	0.52 *	0.48 *	0.58 *	0.46 *	0.76 *	0.58 *	
Mean	30.57	40.26	30.97	10.36	30.8	50.8	30.15	20.64	20.57	50.14	50.43	50.03	10.32	30.76	50.82	20.98	20.62	20.56	50.18	50.32	50.04
Standard deviation	10.00	0.30	0.41	0.72	10.64	0.91	10.52	10.19	10.42	10.03	0.98	10.06	0.51	10.40	0.76	10.45	10.10	10.33	10.01	0.92	10.01

Note. T1 = Time 1; T2 = Time 2; * *p* < 0.05.

**Table 2 ijerph-19-06674-t002:** Fit Indices for the Measurement Models.

Step	Model	χ^2^	df.	CFI	TLI	RMSEA	90%CI RMSEA	AIC	BIC	ABIC
INV1	Configural	1826.991	1043	0.934	0.92	0.053	0.049–0.057	41,334.996	42,886.916	41,740.964
INV2	Metric	1850.939	1067	0.934	0.922	0.052	0.048–0.056	41,325.253	42,773.999	41,704.231
INV3	Scalar	1929.559	1091	0.93	0.919	0.053	0.049–0.057	41,358.178	42,703.749	41,710.167
INV4	Residual variance	1985.453	1127	0.928	0.919	0.053	0.049–0.057	41,414.963	42,605.772	41,726.468
INV5	Residual covariance	2025.773	1139	0.925	0.918	0.054	0.050–0.057	41,441.821	42,581.043	41,739.831
INV6	Factor variance	1981.902	1139	0.929	0.922	0.052	0.048–0.056	41,404.423	42,543.645	41,702.433
INV7	Factor mean	13161.095	1260	0.925	0.917	0.054	0.050–0.057	41,452.982	42,540.616	41,737.497
M1	With free factor loadings	2815.845	1680	0.940	0.930	0.034	0.032–0.036	61,385.800	63,135.500	61,868.800
M2	Stability model	2784.668	1653	0.940	0.930	0.034	0.032–0.036	61,406.843	63,274.963	61,922.519
M3	Structural model	3447.540	1768	0.912	0.902	0.040	0.038–0.042	61,851.070	63,214.865	62,227.600

Note. INV Invariance model step; M Model; CFI = comparative fit index; TLI = Tucker-Lewis index; df = degree of freedom; RMSEA = root-mean-square error of approximation; CI = confidence interval; AIC = Akaike’s information criterion; BIC = Bayesian information criteria; ABIC = sample-size adjusted BIC.

**Table 3 ijerph-19-06674-t003:** Standardized Estimates for the Structural Model: Support from Principals. TPI. and Self-determined Motivation.

Dependant Variables		Estimate (*β*)	S.E.	*p*
	Independent Variables			
T1 Pedagogical expertise				
	T1 Support from principals	0.24	0.05	0.00
T1 Subject matter expertise				
	T1 Support from principals	0.12	0.05	0.02
T2 Amotivation				
	T1 Pedagogical expertise	−0.37	0.24	0.12
	T1 Subject matter expertise	0.10	0.20	0.60
	T1 Amotivation	0.12	0.13	0.35
T2 Controlled motivation				
	T1 Pedagogical expertise	0.15	0.29	0.62
	T1 Subject matter expertise	−0.02	0.26	0.93
	T1 Controlled motivation	0.11	0.07	0.14
T2 Autonomous motivation				
	T1 Pedagogical expertise	0.54	0.31	0.08
	T1 Subject matter expertise	−0.11	0.25	0.66
	T1 Autonomous motivation	−0.04	0.08	0.67

Note. T1 = Time 1; T2 = Time 2; S.E. = Standard error.

**Table 4 ijerph-19-06674-t004:** Standardized Estimates for the Structural Model: Self-determined Motivation. Burnout. and Vigour.

Dependant Variables		Estimate (*β*)	S.E.	*p*
	Independent Variables			
T2 Physical fatigue				
	T2 Amotivation	0.64	0.16	0.00
	T2 Controlled motivation	−0.03	0.07	0.70
	T2 Autonomous motivation	−0.27	0.12	0.02
	T2 Physical fatigue	0.12	0.05	0.01
T2 Cognitive weariness				
	T2 Amotivation	0.39	0.12	0.00
	T2 Controlled motivation	0.01	0.06	0.92
	T2 Autonomous motivation	−0.30	0.10	0.00
	T2 Cognitive weariness	0.30	0.07	0.00
T2 Emotional exhaustion				
	T2 Amotivation	0.40	0.10	0.00
	T2 Controlled motivation	0.00	0.05	0.94
	T2 Autonomous motivation	−0.21	0.09	0.01
	T2 Emotional exhaustion	0.11	0.07	0.08
T2 Physical strength				
	T2 Amotivation	−0.27	0.13	0.04
	T2 Controlled motivation	0.08	0.06	0.18
	T2 Autonomous motivation	0.56	0.09	0.00
	T2 Physical strength	0.12	0.05	0.01
T2 Emotional energy				
	T2 Amotivation	−0.27	0.12	0.02
	T2 Controlled motivation	0.05	0.05	0.36
	T2 Autonomous motivation	0.53	0.09	0.00
	T2 Emotional energy	0.13	0.06	0.02
T2 Cognitive liveliness				
	T2 Amotivation	−0.13	0.10	0.18
	T2 Controlled motivation	0.09	0.05	0.06
	T2 Autonomous motivation	0.62	0.08	0.00
	T2 Cognitive liveliness	0.15	0.05	0.00

Note. T1 = Time 1; T2 = Time 2; S.E. = Standard error.

## Data Availability

The data are available and can be sent by the corresponding authors.

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
