# Peer review of "The Predictive Role of Perceived Support from Principals and Professional Identity on Teachers’ Motivation and Well-Being: A Longitudinal Study"

_ijerph, 2022, doi:10.3390/ijerph19116674_

Round 1

Reviewer 1 Report

I read carefully and with great interest the manuscript. It is built in a rigorous manner, referenced adequately, and offers food for thought, opening new paths in pedagogical research. Having said that, there are some issues that need clarification:

  1. The article relies on two dimensions related to teachers’ motivations and wellbeing: their professional identity, and support from principles respectively. While the first issue received ample support, the latter is very lightly presented. I did not understand from the Results section the specific issues related to the principles offering (or not) the proper support. Also, the article gives little information about the types of support respondents to the questionnaire had in mind. There is a brief information about this in a parenthesis, L 415. I believe the article would gain strength by making the principles-related discussion more visible.
  2. When was the study conducted? Pre-COVID? During the pandemic? Is the socio-economic context of relevance?
  3. In the 4.1 section, Implications for practice, there are some statements that do not derive directly from the research or, at least, the link is not clearly traced. I refer to the idea that teachers need continuous training (L 434), although this is a concept beyond debate. The authors also state that “it appears crucial to alert the whole school community about the risks and benefits related to the principals’ behaviours and management” (L. 437-438). How? Principals behaviours and management are not discussed, only the perceptions of teachers are.
  4. Limits and perspectives section – It is good that authors intend to look into students’ perceptions, but deriving from the article a focus on principals, their leadership skills, behaviours and management style seems more urgent.
  5. The Conclusions section seems underdeveloped, with very broad comments, not highlighting the merits of the study.

Reviewer 2 Report

Dear authors:

The paper is clearly written and soundly supported by theoretical and practical research.

line 156, the word “experience” is not correctly spelled.

Although the main variables involved are widely explained and detailed in the Introduction (vigour, motivation, teacher professional identity and teachers' perceived support from prncipals, there is another term – well-being – whose meaning should be better explained in my opinion.

line 191 and line 354: My suggestion is to add “both” before “perceived support” to better explained the aim of the study

line 409: you say that “to explore the role of the support from principals on TPI” is a central aspect of this study. In my opinion this has not sufficiently highlighted in the Introduction

Reviewer 3 Report

Review Report

The paper under review examines an important question, that of how perceived support from principals and professional identity affects teachers’ motivation and well-being. Although the results are not at all surprising, they are at least informative. The authors have also done a thorough job in combing through the literature and deriving their hypotheses that may help fill some gaps in the literature. The analysis reported is also performed at a much higher level compared to most of the papers published in IJERPH. Below I note a few points that may help improve the paper.

1.       It may be interesting to tie the Introduction to the ongoing COVID-19 pandemic. The pandemic may already have some impact on teachers’ motivation and well-being since the primary instruction method has changed from face-to-face lecturing to online instruction, which may have affected the perceived role of teachers.

2.       About the definitions of teachers. What kind of teachers are you talking about? Secondary school teachers, primary school teachers, or college professors?  

3.       It would be better to explicitly and clearly pose your hypotheses (and number them) in the Introduction section (i.e., Hypothesis 1: …; Hypothesis 2:…), rather than discussing them in the text.

4.       More information on how the participants were recruited should be added for the reader to assess the representativeness of the study sample. What was the sampling frame? Were there refusals? And, how high is the attrition rate? And so on.

5.       It would be informative to explore heterogeneity in the effects of perceived support from principals and professional identity on different teachers. For example, by gender or by levels of school.

6.       There are typos at times. For example, on page 1, line 34: “in another hand” should be “on the other hand”. Also, the bottom of Figure 1: What’s the symbol for “p<0.10”?

Round 2

Reviewer 1 Report

Dear authors,

This is, indeed, a clearer and improved version of the article. A minor issue remained. You state in line 227 that "The research was conducted between September and July 2019". Most probably it is from September 2018 - July 2019 (school year). Other than that, the article is an interesting read.